# Coastal management in Brazil from the perspective of the public policy cycle: Analysis of the "multiscale management" proposed in the National Coastal Management Plan

## Research Article

Brazil; Coastal management; National Coastal Management Plan; multiscale management; public policy; cycle

**Corresponding author:**
Giuliana Pinheiro;
Email: pinheirogiuli@gmail.com

Giuliana Dos Santos Pinheiro[1] (iD), Lise Tupiassu[2] (iD) and
Ana Elizabeth Neirão Reymão[2] (iD)

[1]University Center of the State of Pará, Belém, PA, Brazil and [2]Federal University of Pará and University Center of the State of Pará, Belém, PA, Brazil

## Abstract

This article explores the institutionalization of multiscale management in the Brazilian coastal zone, following the National Coastal Management Plan. It employs Secchi's COQ3 framework to analyze and shed light on this subject. The study uses an exploratory and qualitative approach, employing case studies. It examines the evolution of coastal management in Brazil and highlights the challenges of multiscale management institutionalization. Drawing from Saravia's public policy cycle insights, it assesses coastal management as a public policy, emphasizing variables and deficiencies in the process. The findings indicate fragmentation and regulatory shortcomings. To achieve integrated management, political cooperation is necessary for aligning interests and effective policy implementation across federal scales, contributing to sustainable coastal resource development in Brazil.

## Impact statement

Public policies are designed to address societal needs. Understanding the myriad of terminologies, constraints and legal frameworks applicable to the Brazilian coastal zone within the public policy cycle is of paramount importance. This understanding allows for the recognition that the process of regulating land use and occupancy within this region progresses through various stages, encompassing the identification of societal demands and culminating in the surveillance of policy implementation to assess the effectiveness of the measures taken. Considering the unique characteristics of the Brazilian geographic system, the vast expanse of the coastal zone and its economic significance, it becomes imperative to acknowledge that, despite numerous public policies aimed at reconciling economic growth with environmental sustainability, the integrated management envisaged by the National Coastal Management Plan has yet to be fully realized. Therefore, the present research holds relevance in elucidating that a multiscalar perspective on collaboration among the diverse stakeholders engaged in the development, execution and evaluation of public policies pertaining to the coastal zone is indispensable for effectively addressing the challenges within this area. The perspectives developed throughout this study serve as a catalyst for stimulating discourse within the social, academic and institutional realms, engaging all stakeholders involved in coastal management. These discussions are essential for fostering the integration of the various layers of governance, thus facilitating the formulation of more efficient and effective public policies.

## Introduction

The interdependence between human survival and the preservation of ecosystems has been unequivocally acknowledged (Spinelli et al., 2016). However, the prioritization of economic imperatives has, historically, steered the transformation of landscapes. This economic rationale has underpinned a series of exploitation cycles in the Brazilian coastal territory, from the early Portuguese settlements exploiting pau brasil to the more recent waves of industrialization and tourism development.

Until the 1960s, prevailing political–economic paradigms predominantly revolved around economic development, with the primary objective of standardizing living standards deemed acceptable from a capital-centric perspective. Subsequently, in the following decade, institutional initiatives aimed at addressing environmental concerns and coastal areas began to emerge. One significant development was the establishment of the Interministerial Commission for the

Resources of the Sea (CIRM), which laid the groundwork for a national program of coastal management.

With the advent of postmodernity and the increased recognition of the multifaceted, multiscale nature of sociopolitical, environmental and economic relationships, the concept of territory began to incorporate subjective dimensions. Consequently, the notion of development evolved beyond mere economic growth to encompass these broader dimensions.

In the late 1980s, Brazil initiated the National Program and Plan for Coastal Management, seeking to establish a legal framework for integrated coastal management with the fundamental aim of preserving, conserving and rehabilitating coastal areas. This program encompassed the regulation of activities, the control of land use and the sustainable exploitation of resources within the coastal zone.

One pivotal characteristic of coastal management is the delegation of responsibilities across different levels of governance, including the federal government, states and municipalities. It necessitates synchronized efforts involving the union, 27 states and 247 municipalities (Ministry of the Environment, 2018), alongside numerous governmental and nongovernmental agencies.

Within this intricate framework, multiscalarity is intrinsically tied to integrated coastal management, constituting an essential element of public policies designed to address the challenges arising from the historical exploitation cycles in the region. These policies encompass a constellation of actions, regulations and decisions made by governmental entities and nongovernmental organizations to safeguard fundamental rights, aligning with the priorities defined by the Public Administration (Silva, 2016; Bucci, 2021).

In this context, the primary objective of this research was to investigate the institutionalization of "multiscale management" within the Brazilian coastal zone, as envisioned by the National Coastal Management Plan. To accomplish this, we employ the COQ3 analytical framework proposed by Secchi (2008), which dissects the "How, Where, What, Who and When" variables to expound upon the characteristics of public policies. The study adopts an exploratory qualitative approach, accompanied by the case study methodology, to scrutinize the implementation of "multiscale management" within the Brazilian coastal zone, as articulated in the National Coastal Management Plan.

The research journey commences with a comprehensive review of the historical trajectory and legal frameworks of Brazilian coastal management, illuminating the backdrop against which the model of multiscale management was conceived. Drawing from the theoretical insights of Saravia (2006), the study seeks to comprehend Brazilian coastal management as a manifestation of public policy. Subsequently, the COQ3 scheme, articulated by Secchi (2008) to analyze public policy, is deployed to accentuate the constituent variables of public policy, shedding light on potential shortcomings in the institutionalization process of multiscale coastal zone management.

Finally, the research findings are unveiled, and recommendations are proffered, delineating the requisites for the realization of the integrated, multiscale management envisioned within the National Coastal Management Plan.

## History of Brazilian coastal management

The coastal zone, as the intermediary region between the land and the ocean, encompasses a rich diversity of environments and ecosystems (Crossland and Kremer, 2001). Owing to their unique biophysical characteristics and strategic importance in facilitating communication and transportation, these regions have historically been focal points for human settlement. The chronological account below underscores the economic cycles that have profoundly shaped the exploratory and settlement dynamics of the Brazilian coastal zone:

This historical overview reveals a longstanding pattern of economic exploitation as the central focus of the Brazilian coastal occupation model. Unfortunately, this focus, coupled with limited public investment in urban infrastructure, has led to socio-spatial disparities and environmental degradation in various locales (Frois et al., 2021).

The erratic development of the coastal zone and the unregulated utilization of its resources have had substantial ramifications on the preservation of ecosystems. As a response to the adverse consequences stemming from human activities and the national interest in safeguarding the coastal zone, there arose a need to institute legal mechanisms and specific public policies for land-use planning, occupation control and the conservation of natural resources at the national level. The timeline below elucidates this progression:

Between the 1930s and 1960s, Brazil lacked a coherent environmental policy, instead having sectoral policies concentrated on exploiting natural resources for economic gain. Only in the late 1960s and early 1970s did environmental concerns, including those related to the coastal zone, gain prominence, driven by societal demands concerning industrial pollution.

In 1972, Brazil's participation in the United Nations Conference on the Human Environment in Stockholm precipitated the establishment of the Special Secretariat for the Environment (SEMA). This agency was tasked with devising solutions to regulate industrial and urban pollution. Concurrently, in 1974, the Interministerial Commission for the Resources of the Sea (CIRM) was founded to guide actions associated with marine resource exploitation.

In 1980, the General Guidelines of the National Policy for Sea Resources (PNRM) were enacted, aiming to incorporate the territorial sea, the exclusive economic zone and the continental shelf into the Brazilian territory, with the objective of ensuring the rational and sustainable utilization of marine resources to advance the country's socioeconomic status.

Subsequently, in 1981, Law 6,938 (Brasil, 1981) introduced the National Environmental Policy (PNMA), regulated by Decree 99,274/1990. This policy sought to preserve, enhance and restore environmental quality while reconciling socioeconomic development with environmental protection. The same law established the National Environment System (SISNAMA) and the National Environment Council (CONAMA), with CONAMA serving as the consultative and deliberative body of this system.

The National Coastal Management Program (GERCO), established in 1987 by the Interministerial Commission for Sea Resources (CIRM), laid the foundation for integrated coastal management. Before the enactment of the Federal Constitution of 1988 (Brasil, 1988a), Law No. 7,661 of May 16, 1988 (Brasil, 1988b), established the National Coastal Management Plan (PNGC) with the goal of promoting the sustainable development of coastal regions. The PNGC is an integral part of the National Policy for Sea Resources (PNRM) and the National Policy for the Environment (PNMA), aligning with the principles and objectives of these policies to harmonize economic and social development with the preservation of environmental, historical, ethnic and cultural heritage.

The Federal Constitution of 1988 subsequently guaranteed the right to a balanced environment, emphasizing the significance of coastal and marine ecosystems. In 1992, during the United Nations

**ECONOMIC CYCLES HAVE PLAYED A SIGNIFICANT ROLE IN SHAPING THE EXPLOITATIVE AND OCCUPATIONAL DYNAMICS OF THE BRAZILIAN COASTAL ZONE. THE FOLLOWING INFORMATION IS ADAPTED FROM THE NATIONAL COASTAL MANAGEMENT PLAN: 25 YEARS OF COASTAL MANAGEMENT IN BRAZIL (MMA, 2015).**

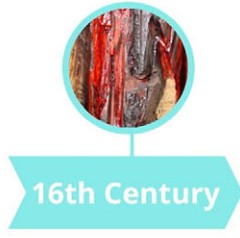 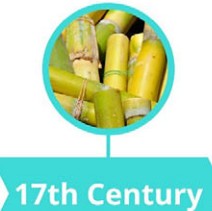 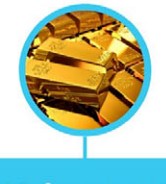 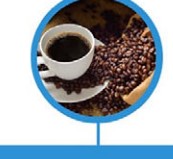 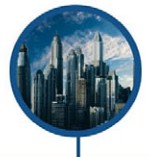

| 16th Century | 17th Century | 18th century | 19th century | 20 th century |
|---|---|---|---|---|
| The initial Portuguese settlements in Brazil were primarily established along the coastal zone, with a prevailing east-west orientation. Within these regions, densely populated areas emerged, including the Bahian Recôncavo, as well as the coastlines of Rio de Janeiro, São Paulo, and the Northeast. Concurrently, communities comprising escaped slaves, indigenous peoples, and traditional populations also appeared in isolated areas (DINIZ, 2008; BORELLI, 2007). It was during this period that the Portuguese extensively exploited Brazilwood | From the second half of the 16th century until the conclusion of the 17th century, the Sesmarias regime allocated public lands to individuals who held political influence. This system served as a precursor to the legal framework of land ownership in Brazil, which does not acknowledge the collective rights of traditional communities regarding land and natural resources (LAMBURGINI, 2018). During this period, the cultivation of sugarcane thrived in coastal regions due to their proximity to areas of sugar exportation. | In the 18th century, Brazil underwent a profound economic shift from coastal sugar production to the "gold cycle." The interior, particularly Minas Gerais, became a focal point as gold was discovered, triggering a surge of settlers and enslaved Africans. This transformative era elevated coastal cities, with Rio de Janeiro at the forefront, as export hubs for gold and precious minerals. The "gold cycle" marked a pivotal juncture in Brazil's history, influencing its socio-economic landscape and contributing to the nation's path toward independence. | In the 19th century, subsequent to the waning prominence of sugar exportation, the Brazilian coastal region underwent anoteworthy metamorphosis, emerging as the principal domain of coffee plantations. | In the 20th century, Brazil's coastal ecosystems bore witness to the forces of industrialization, urbanization, and burgeoning tourism. These ecosystems became intricately enmeshed within industrial complexes and sectors heavily dependent on import-export activities. The emergence of second homes along the coast was propelled by capital influx and governmental investments in the tourism sector, which, in turn, spurred heightened human occupation in environmentally delicate regions, aligning with the broader trend of migration toward urban centers. |

**Figure 1.** Economic cycles that influenced the exploratory and occupational dynamics of the Brazilian coastal zone. Image source: Canva Pro.

Conference on Environment and Development, the need for an integrated approach to comprehend the uses and conflicts in these areas was acknowledged.

This perspective led to the revision of the National Coastal Management Plan (PNGC) through CIRM Resolution No. 5/1997 (Brasil, 1997), establishing the PNGC II as an instrument for integrating government spheres in the development and execution of coastal management. PNGC II remains in effect to this day.

To further advance the integration of public policies within the coastal zone, the Federal Action Plan for the Coastal Zone (PAF) was instituted. The PAF, which underwent revisions and updates in 2005, aims to delineate the Union's activities in these regions. Additionally, Decree No. 5,300/2004 (Brasil, 2004) was enacted to regulate the use and occupation of coastal areas and to delineate the competencies of various levels of government. This decree establishes the limits, objectives and instruments required for coastal management, serving as a complementary measure to the National Coastal Management Plan (PNGC), which did not comprehensively address these aspects.

## Brazilian coastal management and the importance of joint action by federative entities

The principal tools employed for integrated coastal management in Brazil are coastal management plans at the national, state and municipal levels. Nevertheless, numerous environmental management instruments are applied within the coastal zone. Within the framework of integrated coastal management standards for Brazil, nine direct instruments are delineated, as depicted in the following figure:

These instruments serve the purpose of defining responsibilities and institutional procedures, demarcating territorial usage, amassing socio-environmental data, evaluating the efficacy of management initiatives and guiding coastal management endeavors. Nevertheless, the intricacies of this subject necessitate the use of various management instruments that are not explicitly addressed by Decree No. 5,300/04 or the National Coastal Management Plan. While not exclusively dedicated to integrated coastal management, these instruments significantly influence territorial planning and biodiversity preservation in the coastal region. Illustrative examples

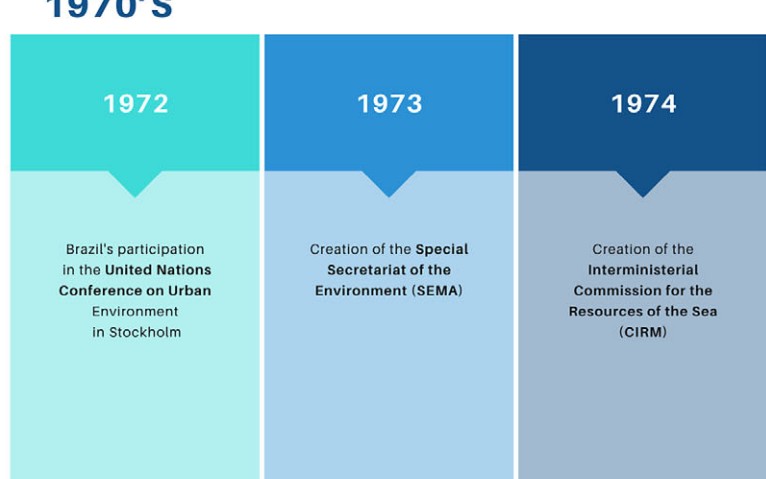

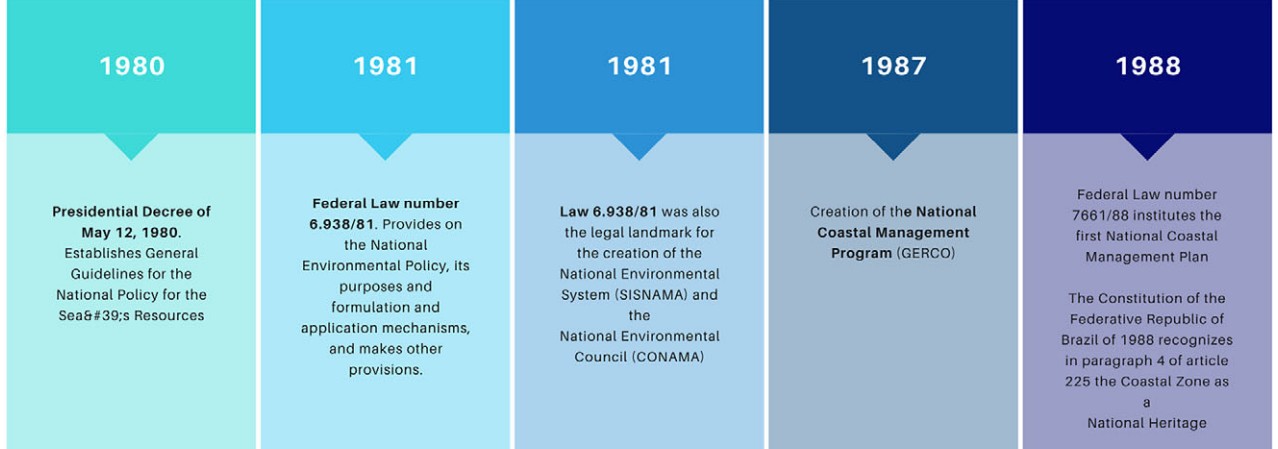

**Figure 2.** Main legal frameworks of Brazilian coastal management.
Source: Prepared by the authors.

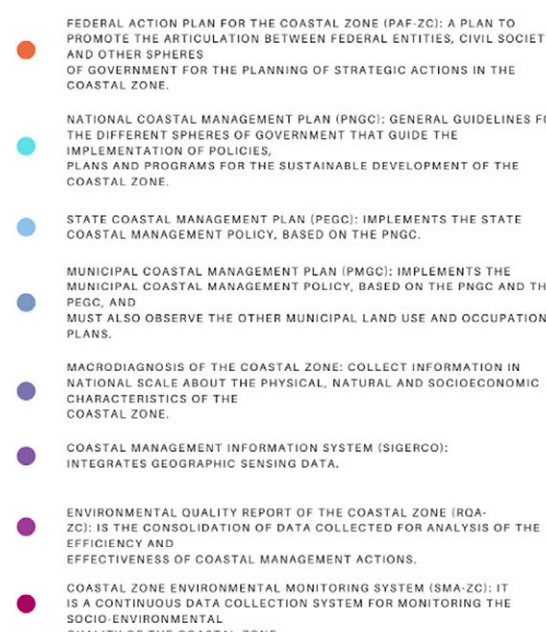

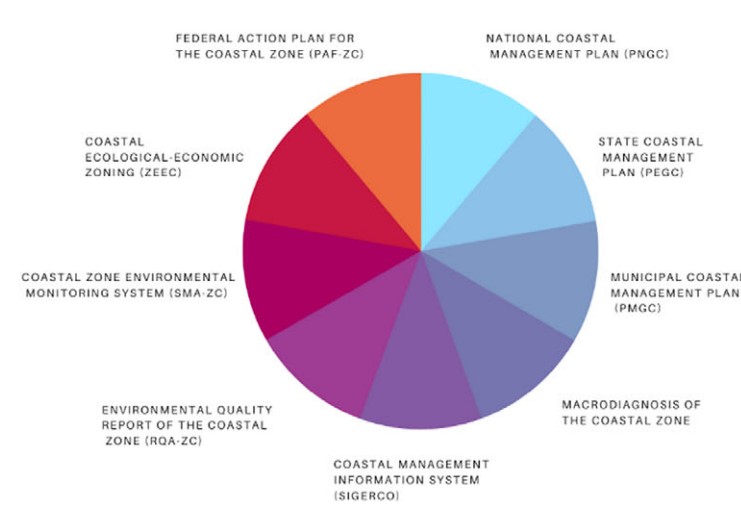

**Figure 3.** Instruments pertaining to Brazilian coastal management according to Decree No. 5,300/2004.
Source: Prepared by the authors.

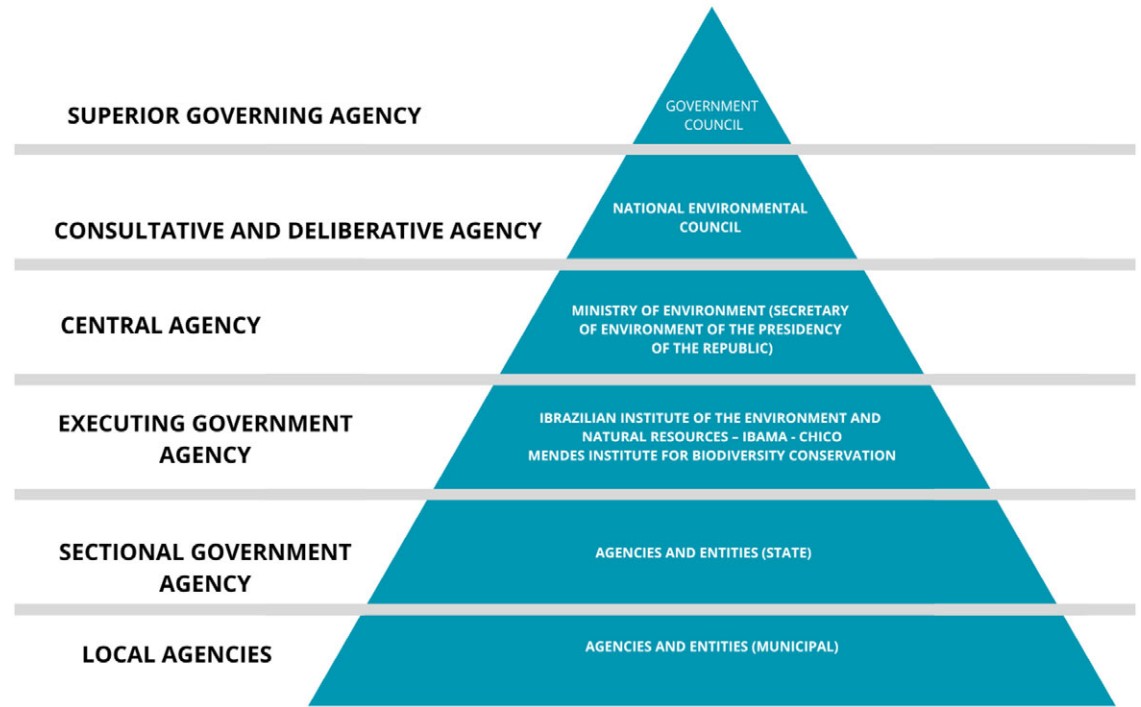

**Figure 4.** Structure of the National Environment System.
Source: Prepared by the authors.

of such instruments include the Water Resources Plans (Law No. 9,433/97), Solid Waste Plans (Law No. 12,305/10), Basic Sanitation Plans (Law No. 11,445/07) and Municipal Master Plans (Law No. 10,257/01 – City Statute).

Integrated coastal management stands as a fundamental tenet within the National Coastal Management Plan and constitutes a cornerstone of national environmental governance. The National Environment System (SISNAMA), as stipulated by Law No. 6,938/1981 and regulated by Decree No. 99,274/1990, shoulders the responsibility for environmental governance throughout Brazil, inclusive of the coastal zone. The composition of SISNAMA is delineated in the subsequent figure:

Within SISNAMA, we encounter the National Council for the Environment (CONAMA), an advisory and decision-making body endowed with the authority to formulate guidelines and standards pertaining to the implementation of the PNGC (National Coastal Management Plan) and other environmental policies. Furthermore, the Ministry of the Environment (MMA) serves as the central authority charged with orchestrating and coordinating environmental initiatives, whereas the Brazilian Institute for the Environment and Renewable Natural Resources (IBAMA) and the Chico Mendes Institute for Biodiversity Conservation (ICMBio) act as the executive arms responsible for executing national environmental policies (Ministry of Environment Affairs 2019). In addition to these entities, SISNAMA encompasses agencies and bodies within the federal, state and municipal public administration, vested with the mandate of safeguarding environmental quality, devising resource utilization strategies and overseeing and inspecting activities contributing to environmental degradation (Asmus et al., 2006).

From the aforementioned elucidation, one may deduce that the regulation of coastal zone use and occupation necessitates a harmonious collaboration between the federal government, states and municipalities, with the overriding objective of conserving the natural, historical, biological and cultural richness of the coastal zone while accommodating economic pursuits (Silva and Etges, 2019). This multitiered approach has engendered the formulation and implementation of comprehensive public policies in the coastal domain.

Coastal management in Brazil hinges on the federal compact uniting the union, states and municipalities. At the national echelon, the National Policy for Marine Resources, the National Policy for the Environment and the National Plan for Coastal Management have been enacted. The National Coastal Management Plan incorporates federal legislation governing land-use planning and territorial organization, directives from State Management and Zoning Plans and municipal standards governing urban land utilization and development. At the state and municipal levels, the coordination of coastal management falls under the purview of entities such as state and municipal environmental, urban and territorial planning agencies.

On a national scale, this management paradigm draws its underpinnings from the National Policy for Marine Resources, the National Policy for the Environment and the National Plan for Coastal Management. As stipulated by the provisions of the National Coastal Management Plan, consideration must also be accorded to federal statutes governing land-use planning and territorial organization, the principles articulated within State Management and Zoning Plans and the local regulations pertaining to urban land utilization and development. Therefore, at the state and municipal tiers, various institutions, ranging from state and municipal environmental entities to urban and territorial planning agencies, shoulder the responsibility for coordinating coastal management initiatives.

Hence, it falls to the union to oversee the implementation of the National Coastal Management Plan, coordinating, supervising, incentivizing and monitoring decentralized undertakings at the federal level. States are tasked with conceiving, executing, implementing and overseeing the State Coastal Management Plan, instituting the state information system, advancing monitoring programs and fostering intersectoral and interinstitutional cooperation within their purview (Barragán, 1998). Meanwhile, municipalities are charged with incorporating the proposals stemming from the state coastal management program into the Municipal Coastal Management Plan, ensuring that municipal master plans for land occupation conform to the national and state coastal

management paradigms. Moreover, municipalities must establish municipal coastal management information systems and institute and oversee coastal monitoring programs (Barragán, 1998).

Brazilian coastal management hinges on territorial policies that govern coastal land use by an array of actors and sectors and necessitates their collective collaboration with federal entities (Ruckert, 2016). Within this framework, multiscale cooperation should be comprehended as an institutional structure established by individuals to realize common objectives in compliance with the law and the legal framework (Saquet, 2010). Nevertheless, the intent of political agents stands as a pivotal prerequisite for the actualization of this collaborative framework. A lack of comprehension regarding existing relationships and processes stands as one of the obstacles hampering development, resulting in fragmented activities and a dearth of community participation (Saquet, 2010).

Multiscale coastal management in Brazil confronts challenges occasioned by the diversity of stakeholders and instruments involved. The institutionalization of this approach exhibits variations across coastal regions of the nation, replete with impediments that hinder collaboration between coastal states and municipalities. Disparities in local policies, funding, technical capacity and social organization bear upon the compatibility of public policies at different federal levels (Asmus et al., 2006).

In this context, there arises a compelling imperative for in-depth investigations into the processes underpinning coastal management, geared toward the formulation of regionally nuanced public policies capable of fostering sustainable development within the region via integrated political and social governance.

## Coastal management as public policy: A perspective through the COQ3 method

In-depth qualitative research was conducted to enhance our comprehension of the institutionalization of multiscale management within the framework of the Brazilian coastal zone, as envisioned in the National Coastal Management Plan. This exploratory analysis offers a comprehensive view of the subject, identifying patterns and ideas, all without testing specific hypotheses (Collis and Hussey, 2005). Utilizing a qualitative approach enables us to gain a deeper insight into the intricacies and contextual factors at play in the institutionalization of multiscale management as a public policy (Minayo, 2003).

The research employed a case study methodology to investigate the institutionalization of "multiscale management" within the Brazilian coastal zone, as articulated in the National Coastal Management Plan. The case study approach is particularly apt for delving into dynamic scenarios, providing answers to the "how" and "what" questions (Yin, 2001).

After determining the subject of the case study, a chronological analysis of Brazilian coastal management was conducted, with a primary focus on the integrated multiscale perspective. This analysis took into account legal frameworks and economic cycles. Throughout this process, coastal management was regarded as a facet of public policy, and we applied the COQ3 method, as proposed by Secchi (2008), to scrutinize the constituent elements of multiscale management within the Brazilian coastal zone. We endeavored to pinpoint shortcomings through the assessment of the variables outlined below:

The findings of this analysis underscore the imperative to adopt an input-process-output-impact framework in the planning and execution of coastal management endeavors. It is vital to diligently

monitor and assess public policies across all federal tiers, factoring in the unique attributes of coastal regions. This holistic approach is paramount in our commitment to preserving ecosystems, fostering economic growth and enhancing the quality of life for local communities.

### Coastal management as a public policy: The application of the COQ3 method in analyzing the institutionalization of multiscale coastal management proposed in the PNGC

Public policies represent governmental actions that follow a structured cycle, involving various actors, focal points and processes. According to Saravia (2006), the creation of public policies encompasses the following stages: agenda setting, elaboration, formulation and implementation. The agenda setting phase pertains to the inclusion of societal demands within governmental priorities. During the elaboration stage, issues are identified and defined, and alternative solutions are assessed. The formulation phase involves the selection of an alternative solution and the establishment of objectives and legal frameworks. The implementation stage encompasses resource planning, organizational strategies and the development of plans, programs and projects through which the policy becomes operational. Subsequently, the policy enters the execution phase, wherein it is put into action. The monitoring and evaluation phases entail oversight and the analysis of policy outcomes.

To gain a more profound understanding of the institutionalization of multiscale coastal management, as envisaged in the National Coastal Management Plan, this study employs the COQ3 method.

In analyzing the "How" variable, it becomes apparent that coastal management policy in Brazil has primarily been characterized as reactive, employing "command and control" approaches to mitigate adverse impacts (Almeida, 1997; Secchi, 2008). The "Where" variable underscores the federal nature of the nation, with coastal management responsibilities shared among federative entities (Silveira, 2017). With respect to the "What" variable, it is evident that regulatory public policies are influenced by political considerations, and the chosen solutions are not always the most suitable (Secchi, 2008, 2010). As for the "Who" variable, the degree of participation and influence of political actors hinges on the strength of their involvement and the ramifications of policies on their activities (Secchi, 2010). In this context, coastal management necessitates political coordination and a comprehensive examination that takes into account natural, historical and cultural dimensions (Cicin-Sain, 1993).

In the realm of Brazilian coastal management, the policies implemented by diverse entities lack a clear hierarchy of importance, leading to potential overlaps and conflicts between policies due to the jurisdictional competencies of federative entities and the interests of the stakeholders involved. Given the diverse interests of these stakeholders, it is essential to align their objectives through procedural accountability, wherein agents are subject to oversight during the decision-making process, as well as performance accountability, involving diverse and horizontal metrics for the assessment of public agencies (Bevir, 2011). It is noteworthy that indicators and metrics play pivotal roles in the social construction of reality, even though they may not capture the entirety of the complex real-world scenarios (Telles, 2003).

By means of a documentary analysis of the report titled "25 Years of Coastal Management in Brazil," Cavalcante and Aloufa (2018) have discerned that legal instruments, plans, programs and projects have yielded positive outcomes in coastal management. Nevertheless, there remain lacunae that necessitate addressing for the effective management of natural resources and coastal occupancy.

In this context, the non-integrated efforts of agents and institutions perpetuate challenges that have long been subjects of academic and governmental deliberation. These issues encompass, among others, oil spills, unregulated urbanization leading to the impermeabilization of coastal areas due to irregular developments, erosion, the depletion of mangrove and restinga vegetation, land-use disputes, water contamination, impacts on fisheries, dredging, noise pollution, air pollutant emissions and the generation of solid waste (Scherer et al., 2010; Figueiredo, 2013; Ministry of the Environment, 2015; Cavalcante and Aloufa, 2018).

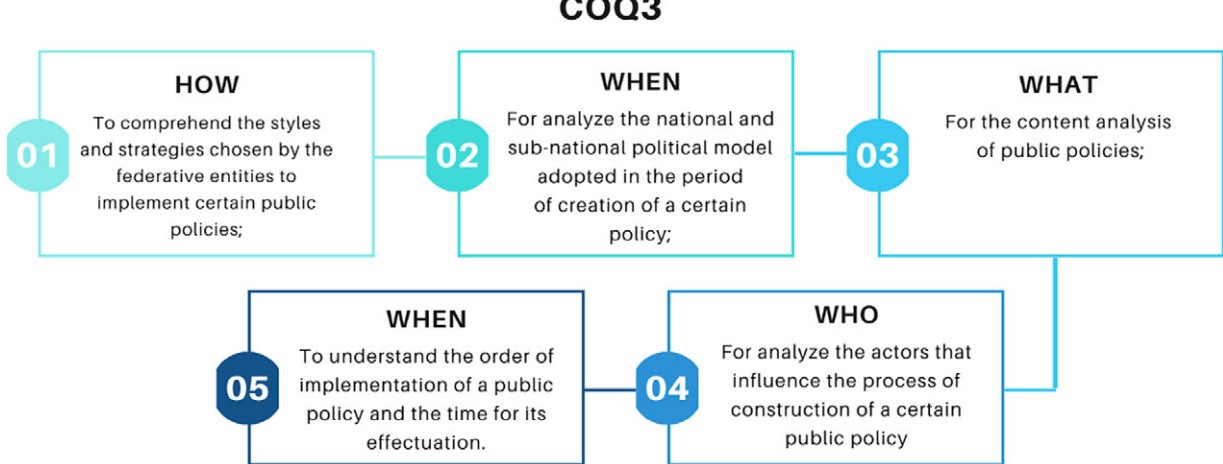

**Figure 5.** Analytical variables of the COQ3 scheme.
Source: Prepared by the authors.

## Research findings

The application of the COQ3 method to analyze Brazilian coastal management policies underscores the imperative need to fortify coordination mechanisms and evaluate existing public policies. The focal point of these efforts is the integration envisioned in the National Coastal Management Plan, all with the overarching goal of promoting the sustainable development of the coastal zone. An examination of coastal management instruments reveals the intricate interplay among federal, state and municipal levels through planning, evaluation and monitoring tools. Nevertheless, research indicates that the implementation of these instruments has often been haphazard across the national coastal zone.

On the federal front, the Federal Action Plan for the Coastal Zone (PAF-ZC) holds significance. This policy strives to establish prior coordination among federal entities, civil society and other governmental spheres for the planning of strategic actions and the integration of public policies in the coastal zone. In its 2015–2016 iteration, the PAF-ZC included the promotion of forums for sectoral plan discussion and integration. Regrettably, only 25% of the sixteen planned actions were executed within the stipulated timeframe (Coelho, 2017). The PAF-ZC 2017–2019 outlined eighteen strategic actions in priority areas, with a particular emphasis on civil society participation in some, such as Action Plan 15. The goal of this plan is to ensure civil society's involvement in the decision-making processes pertaining to coastal and marine management in Brazil, thereby contributing to the realization of Sustainable Development Goal 14 (Life below Water) in the country (Souto, 2020).

At the state level, the establishment and execution of State Coastal Management Plans and Coastal Ecological-Economic Zoning have transpired with varying motivations, leading to disparities among coastal states. Up until 2013, only eight of the seventeen coastal states had initiated Coastal Ecological-Economic Zoning, and merely five had crafted a State Coastal Management Plan. In 2020, the State of Pará introduced the State Policy for Coastal Management, which encompasses 47 coastal municipalities and relies on municipal support for the implementation of its instruments (Semas/PA, 2022).

In the realm of Municipal Coastal Management Plans, their implementation has been found to be lacking. Despite Brazil having 274 coastal municipalities, only a handful have established Municipal Coastal Management Plans. Examples include the municipality of Florianópolis in Santa Catarina, with the PMGC created by Law No. 7975/2009; the municipality of Campos de Goytacazes in Rio de Janeiro, with the PMCG established by Law No. 8335/2013; and the municipality of Anchieta in Espírito Santo, with the PMGC created by Law No. 973/2014.

The management instruments envisaged in the National Coastal Management Plan have not yielded the desired success. The implementation of coastal management at multiple scales has proven to be complex and relatively inefficient. State and municipal instruments for managing the area are still in the nascent stages of development.

Consequently, it is vital to engage in monitoring and evaluation of public management policies at all federal levels throughout the planning and implementation cycles. These actions enable a thorough assessment of the potentialities and vulnerabilities of coastal areas, taking into account their physical characteristics and patterns of occupancy. The overarching goals of these policies are to preserve, conserve and rehabilitate coastal ecosystems while maintaining their economic potential and enhancing the quality of life for coastal communities, as enshrined in the objectives of the PNGC.

In the context of institutional planning, although the creation of management instruments has advanced the protection of the coastal zone, addressing issues related to these policies necessitates long-term environmental monitoring by all relevant entities. This includes the effective implementation of State Coastal Management Plans, as well as municipal policies governing coastal management and urban development. Furthermore, continuous environmental education is essential to foster societal engagement in the formulation of public policies and disseminate knowledge about the intricacies of the national coastal zone.

Ultimately, the realization of integrated and multiscale coastal management hinges on the harmonization of plans, projects and standards among the union, states and municipalities. This entails a comprehensive consideration of all sectoral activities conducted in the coastal zone, along with the socioeconomic and environmental challenges encountered, with the ultimate objective of devising coherent policies across all federative levels.

## Final considerations

The examination of the institutionalization of "multiscale management" in the Brazilian coastal zone, as outlined in the National Coastal Management Plan and analyzed through Secchi's method (Secchi, HYPERLINK \l "B29"2008), reveals that coastal management, operating as a "command and control" policy, possesses the capability to impose constraints on polluting agents and is subject to influence by various interest groups. In this intricate landscape, coastal management in Brazil confronts challenges stemming from the division of competencies among federative entities. This division can hinder integration and the realization of integrated coastal management, given that each level of government harbors its unique responsibilities and interests. The absence of institutional and regulatory coherence jeopardizes the efficacy of coastal management, necessitating a reevaluation of public policies designed for the coastal zone.

Although not explicitly articulated in legal documents, the multi-scale perspective is implicit within the integrated management framework proposed in the National Coastal Management Plan. This framework endeavors to harmonize the diversity of legislation and the political-administrative organization to align with the interests established within the coastal zone. For this management model to be effectively implemented, it is imperative to recognize that coastal management is an institutional, participatory, adaptable and consequently a dynamic and ongoing process. It comprises sectoral, spatial and interdisciplinary public policies aimed at the holistic development of the coastal zone, enhancing the quality of life of its human inhabitants while concurrently preserving the biological diversity, productivity and economic potential of the coast.

**Open peer review.** To view the open peer review materials for this article, please visit http://doi.org/10.1017/cft.2023.28.

**Data availability statement.** All datasets used for the analysis are publicly available at the corresponding references.

**Acknowledgements.** The authors extend our heartfelt gratitude to all family members, friends and fellow researchers who, whether directly or indirectly, lent their support to this endeavor. We remain optimistic that the present research will serve as a catalyst for further investigations into multiscale management within the coastal zone, thereby fostering a heightened awareness of the imperative need for coordination among the various federative levels engaged in the stewardship of the Brazilian coastal zone. Such coordination is fundamental to

the realization of holistic and constructive environmental and public policies, encompassing economic, environmental and social dimensions.

**Author contribution.** The composition of this manuscript was the collaborative effort of Giuliana Pinheiro, Lise Tupiassu and Ana Elizabeth Reymão. The translation of the text into English was undertaken by American Journal Experts, with the invaluable assistance of Akira Takatsuji in the creation of the figures.

**Financial support.** It is imperative to note that this research was conducted without the receipt of any dedicated grant or financial support from either public funding agencies or private enterprises, whether for-profit or not-for-profit in nature.

**Competing interest.** The authors declare none.

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
