## [Reviewer Report]

This manuscript contributes positively to discussions on coastal zone management policies, providing an overview of Brazil’s experience. It provides a good opportunity for experience exchange between specialists in the field and all those interested on this complex topic, coming from different backgrounds and/or nationalities.

As mentioned several times throughout the manuscript, the central theme of this work is of a complex nature because it involves a variety of multiscale policies, stakeholders and issues over a long period of time. Management, regulation and protection of the coastal zone are of paramount importance, given that we are on the second year of the UN Ocean Decade and that coastlines are at the frontline of climate change. There is no doubt that, globally, coastlines and coastal zones require very special attention when it comes to the implementation of dedicated policies that support use regulations, sustainable provision and protection of coastal assets, ecosystem services and resources.

This manuscript outlines Brazil’s journey from the creation to the implementation of multiscale policies that were proposed in the National Coastal Management Plan. However, there is a lot of information described in the text about the timeline of events, their purposes and the organisations involved, making the understanding of the message being conveyed very difficult.

From my perspective, after reading this manuscript three times, I do not think that it has achieved its objective, defined as: “to analyze whether the integrated ‘multiscale management’ of the Brazilian Coastal Zone’, objectified by the National Coastal Management Plan, is effectively carried out”. The answer to this question is definitely within this manuscript, but it needs, in my perception, to be better structured so that it is clearly stated and perceived.

I suggest the authors to reframe this manuscript to better explain their objective, why it is important, how they implemented the research (clearer method approach needed) and emphasise their findings. I believe that all the necessary information is already within the text, but it would benefit from restructuring.

For instance, the authors could outline the main policies that are relevant to the National Coastal Management Plan in the form of a timeline, indicating what policies were created by whom and when.

When discussing multiscale policy making and management, the authors could create figures/ organograms to summarise and indicate the direction of policy flow, including organisations and their roles (e.g., MMA, IBAMA, ICMBio, etc.).

It might be a good idea to consider restructuring the current manuscript in the form of a case study, providing background information with relevant definitions and need for such investigation, clear steps of the methodology used, followed by the presentation of your main findings, limitations to investigation approach, and conclusion.

I hope my comments will not discourage the authors to resubmit this important piece to “Coastal Futures”, as I think the information here is very good. This work does require tweaking in order to be published, but as mentioned before, the information is already within the text. What is needed here is restructuring the text, making use of figures to summarise and clearly convey information, so that we – readers – can identify the thought process behind this work, critical appraisal of information and findings.

I congratulate the authors on this comprehensive piece, wishing them my best.

Thank you very much.

Aline

---

## [Reviewer Report]

The idea of the paper is excellent, but the manuscript lacks of structure, up-to-date literature review, clarity and sound science.

---

## [Reviewer Report]

Dear authors,

I have reviewed your manuscript entitled “The coastal management in Brazil from the perspective of the public policy cycle: an analysis of the “multiscale management” proposed in the National Plan for Coastal Management” and have some feedback to share with you.

Overall, the manuscript provides a good and detailed narrative of the creation and development of integrated coastal zone management in Brazil, which presents a unique perspective on coastal management and policy. It highlights the complexity and difficulties that large countries like Brazil face in integrating actions to manage, let alone protect, natural coastal assets. The manuscript’s contribution to coastal and marine policy, focused on a national context but useful as a global reference in the field, is commendable. Given the scarcity of studies investigating the effectiveness of coastal management and policy strategies outside of the topic of protected areas, this work is especially valuable.

However, the manuscript is long and at times repetitive, although this does not compromise its overall quality. One area for improvement would be to include examples of where multiscale approaches have worked in practice, even if only partially. Highlighting positive aspects of such cases could provide useful reference points for future action. However, the authors may consider this to be outside the scope of their current work. It would also be helpful to organise the findings in a table to facilitate the appreciation of the results of the variables used (“How”, “Where”, “What”, “Who”, and “When”), but it is ultimately up to the authors to decide whether or not to implement this structural change. These are only suggestions.

The choice of tense used throughout the manuscript should be consistent, and the minor errors identified below should be corrected:

• Line 147 - *gained

• Line 179 present tense - the whole text should either be in present or past tense.

• Lines 331 and 533: “created by men”, or by people?

• Figure 1: would benefit from an associated timeline corresponding to the cycles.

• Figure 4: there is a typo in “Brazilian Institute of the Environment...”

Overall, I find this manuscript to be a valuable contribution to the journal and the field of coastal and marine policy, and I recommend it for publication.